# AUGMENTING REPRESENTATIONS WITH SCIENTIFIC PAPERS

**Nicolò Oreste Pinciroli Vago**
Department of Electronics, Information and Bioengineering, Politecnico di Milano
Via Giuseppe Ponzio, 34, Milan, 20133, MI, Italy
Osservatorio Astronomico di Roma, INAF
Via Frascati 33, Monte Porzio Catone, 00078, RM, Italy
AstroAI, Center for Astrophysics | Harvard & Smithsonian
60 Garden Street, Cambridge, 02138, MA, USA
`nicolooreste.pinciroli@polimi.it`

**Rocco Di Tella**
AstroAI, Center for Astrophysics | Harvard & Smithsonian
60 Garden Street, Cambridge, 02138, MA, USA

**Carolina Cuesta-Lázaro**
AstroAI, Center for Astrophysics | Harvard & Smithsonian
60 Garden Street, Cambridge, 02138, MA, USA
Center for Computational Astrophysics, The Flatiron Institute
162 5th Ave 9th floor, New York, 10010, NY, USA

**Michael J. Smith**
AstroAI, Center for Astrophysics | Harvard & Smithsonian
60 Garden Street, Cambridge, 02138, MA, USA

**Cecilia Garraffo**
AstroAI, Center for Astrophysics | Harvard & Smithsonian
60 Garden Street, Cambridge, 02138, MA, USA

**Rafael Martínez-Galarza**
AstroAI, Center for Astrophysics | Harvard & Smithsonian
60 Garden Street, Cambridge, 02138, MA, USA

## ABSTRACT

Astronomers have acquired vast repositories of multimodal data, including images, spectra, and time series, complemented by decades of literature that describes the physics of associated astrophysical sources. Still, both data modalities (numerical data and text) are rarely systematically integrated. This work introduces a contrastive learning framework that aligns astrophysical X-ray spectra with domain knowledge extracted from the scientific literature, thereby facilitating the development of shared multimodal representations. Establishing this connection is inherently complex, as scientific texts encompass a broader and more diverse physical context than spectra. We propose a contrastive pipeline that achieves a 20% Recall@1% when retrieving texts from spectra, proving that a meaningful alignment between these modalities is not only possible but capable of accelerating the interpretation of rare or poorly understood sources. Furthermore, the resulting shared latent space effectively encodes physically significant information. By fusing spectral and textual representations, we improve the estimation of 20 physical variables by $16 - 18\%$ over unimodal spectral baselines. Our results indicate that a Mixture of Experts (MoE) strategy, which leverages

both unimodal and shared representations, yields superior performance. Finally, outlier detection within the multimodal latent space identifies high-priority targets for follow-up investigation, including a candidate pulsating ULX (PULX) and a gravitational lens system. Importantly, this framework can be extended to other scientific domains where aligning observational data with existing literature is possible.

# 1 INTRODUCTION

Foundation models are large-scale neural networks pre-trained on diverse data and adaptable to multiple downstream tasks, and have recently been applied to astronomy (Parker et al., 2024; Leung & Bovy, 2024). In astronomy, upcoming surveys (Vera Rubin Observatory, Roman Space Telescope) will generate petabyte-scale multimodal datasets (Greenstreet et al., 2024; Hernandez et al., 2024; Gezari et al., 2022), which need scalable approaches to extract scientific insights. Unlike single-modality foundation models, astronomical observations are inherently multimodal: a single source may have images, spectra, light curves, and decades of textual descriptions in scientific literature, each capturing complementary physical information.

There exist both unimodal and multimodal astronomical foundation models (Parker et al., 2024; Mishra-Sharma et al., 2024; Rizhko & Bloom, 2025). However, the systematic integration of observational data with textual scientific knowledge remains unexplored. This gap is significant, as scientific literature contains high-quality peer-reviewed expert interpretations, physical models, and contextual information unavailable in raw observations alone.

We present the first contrastive learning framework aligning X-ray spectra with scientific papers' summaries, creating a shared latent space that enhances spectral data and encodes physical properties. This approach addresses key challenges for astronomical foundation models by fusing heterogeneous data modalities such as spectra and text, preserving physically meaningful structures in learned representations, and enabling knowledge transfer across observational datasets and scientific literature.

The contributions of this work can be summarized as follows: (1) alignment of X-ray spectra with textual summaries using contrastive learning; (2) demonstration that multimodal representations outperform unimodal ones for physical parameter estimation (3) 97% data compression (4,672 to 128 dimensions) while preserving relevant physical information; and (4) ability to use the enriched latent space to flag outliers.

# 2 METHODS

## 2.1 DATASET

An X-ray spectrum is a measurement of the distribution of detected photons across a range of energy channels. For Chandra observations (Evans et al., 2024), a spectrum is derived from individual photon events with energies between 0.5 and 8 keV. In this work, for each X-ray detection, we discretize the spectrum into 400 energy bins where the intensity is measured as a photon count rate (photons per unit of time and energy). These data are min-max normalized to allow the model to learn from the relative distribution of energy (the spectral shape), which serves as a physical signature of the source's underlying processes (see also Pinciroli Vago et al. (2025b)).

To incorporate expert knowledge, we cross-reference the spectral data with the NASA Astrophysics Data System (ADS), which contains a collection of scientific papers in astronomy and astrophysics. Cross-referencing is allowed by the use of sky coordinates and source identifiers from SIMBAD.

The final dataset consists of 11,447 spectrum-text pairs, split into training (69%), calibration (1%), validation (15%), and test (15%) sets. Each sample is associated with up to 20 ground-truth physical variables (see Table 1) used to evaluate the physical consistency of the learned representations.

Table 1: Physical variables and their descriptions from the Chandra Source Catalog. These 20 variables serve as the ground truth for evaluating the physical interpretability of the learned latent representations and are used in the multimodal regression tasks.

| Variable | Description |
| --- | --- |
| hard_hs | Hardness ratio between hard (2.0–7.0 keV) and soft (0.5–1.2 keV) bands |
| hard_ms | Hardness ratio between medium (1.2–2.0 keV) and soft (0.5–1.2 keV) bands |
| hard_hm | Hardness ratio between hard (2.0–7.0 keV) and medium (1.2–2.0 keV) bands |
| var_prob_b | Gregory–Loredo variability probability |
| var_index_b | Intra-observation Gregory-Loredo variability index |
| powlaw_gamma | Photon index from power-law spectral fit |
| powlaw_nh | Hydrogen column density from power-law fit |
| powlaw_stat | Fit statistic for the power-law spectral model |
| bb_kt | Temperature (keV) from blackbody spectral fit |
| bb_nh | Hydrogen column density from blackbody fit |
| bb_stat | Fit statistic for the blackbody model |
| brems_kt | Temperature (keV) from Bremsstrahlung fit |
| brems_nh | Hydrogen column density from bremsstrahlung fit |
| brems_stat | Fit statistic for the bremsstrahlung model |
| apec_kt | Temperature (keV) in APEC thermal model fit |
| apec_abund | Abundance of the best fitting absorbed APEC model spectrum to the source region aperture Pulse Invariant spectrum |
| apec_z | Redshift in APEC fit |
| apec_nh | Hydrogen column density from APEC fit |
| apec_stat | Fit statistic for the APEC model |
| flux_significance_b | Flux significance |

## 2.2 ARCHITECTURE

Our pipeline, presented in Figure 1, follows foundation model principles: we use two pre-trained unimodal foundation models followed by contrastive alignment, focusing the evaluation on downstream tasks.

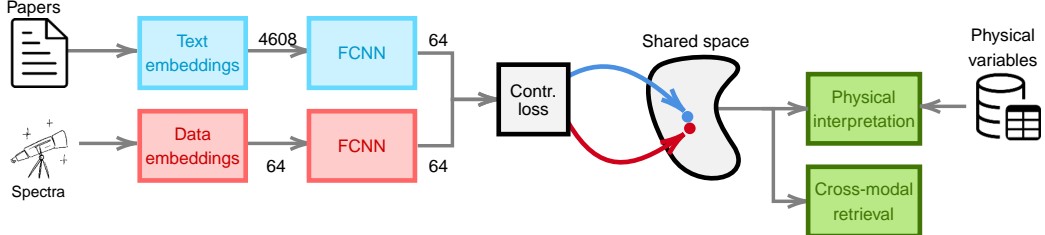

Figure 1: Pipeline overview. Spectra are encoded via a transformer-based autoencoder. Scientific papers are summarized using GPT-4o-mini, and the summaries are embedded from OpenAI's Ada-002. Contrastive learning aligns modalities into a shared latent space for downstream tasks.

The spectra are processed by the transformer-based autoencoder introduced in Pinciroli Vago et al. (2025b). In this work, we compress spectra to 64-dimensional latent vectors, minimizing the reconstruction loss (MAE). The textual summaries are generated from scientific papers using GPT-4o-mini and later embedded using OpenAI's Ada-002 model. A subset of these GPT-generated summaries was manually verified by expert astrophysicists for scientific consistency with the source papers.

Two fully-connected networks map spectral (64-dimensional) and text (4,608-dimensional) embeddings to a shared 64-dimensional space. We optimize the InfoNCE loss (Oord et al., 2019):

$$\mathcal{L}_{\text{InfoNCE}} = -\frac{1}{N} \sum_{i=1}^{N} \log \frac{\exp(\text{sim}(t_i, d_i)/\tau)}{\sum_{j=1}^{N} \exp(\text{sim}(t_i, d_j)/\tau)} \tag{1}$$

where $\text{sim}(x,y)$ is cosine similarity, $\tau$ is temperature (tuned on calibration set), and $(t_i, d_i)$ are matched text-data pairs.

For downstream tasks, we explicitly define three representation settings. Let $z_s \in \mathbb{R}^{64}$ and $z_t \in \mathbb{R}^{64}$ be the aligned spectral and text embeddings after contrastive training. "Spectra" uses only $z_s$, "Text" uses only $z_t$, and "Both" uses early fusion by concatenation,

$$z_{\text{both}} = [z_s; z_t] \in \mathbb{R}^{128}, \tag{2}$$

We also evaluate the results on three downstream tasks:

- Cross-modal retrieval: we retrieve text descriptions from spectra using similarity search.
- Physical parameter regression: we use a $k$-NN regressor (with $k = 3$) to predict 20 physical variables from latent representations. For the multimodal setting ("Both"), we use the concatenated vector $z_{\text{both}}$. We employ a Mixture of Experts (MoE) strategy: for each variable, we select the best representation (pre- or post-alignment, using texts, spectra, or both) based on validation set Pearson correlation.
- Outlier detection: we use Isolation Forest to identify rare sources in the aligned latent space.

## 2.3 TRAINING AND EVALUATION

Models are trained with Adam optimizer, performing a grid search in the hyperparameter space over learning rate ($10^{-4}$ to $10^{-3}$), shared space dimension (16 to 128), dropout (0.1 to 0.5), and hidden dimensions (16 to 1024). The performance metrics are Recall@k% (i.e., the proportion of queries with correct match in top $k$%), Median Rank, MAE (for regression), and Pearson correlation (for latent space-physical variables relationships).

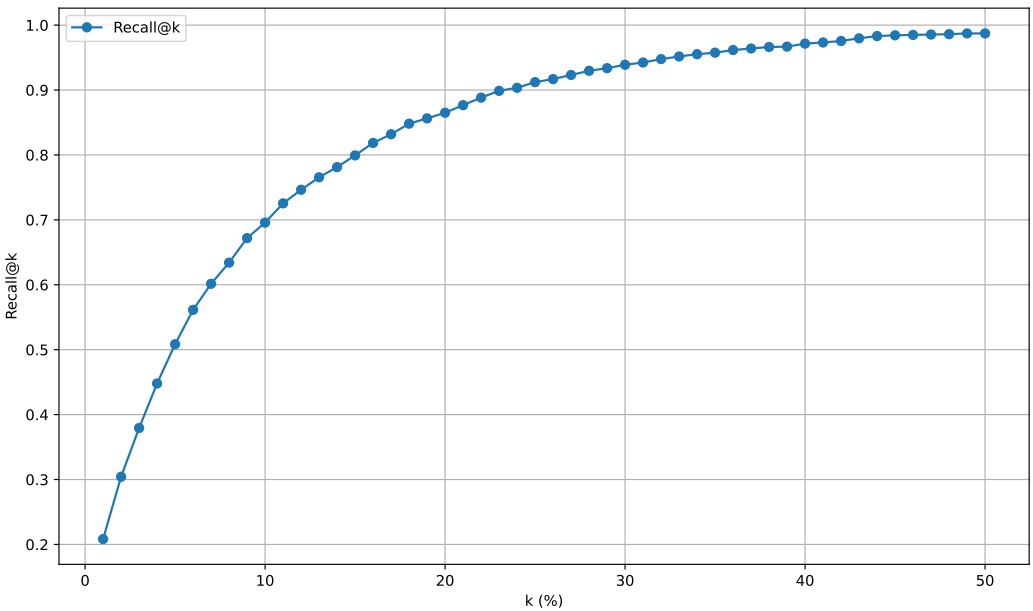

Figure 2: Recall@k% as a function of $k$, expressed as a percentage of the test set, for the ensemble model.

## 3 RESULTS

### 3.1 CROSS-MODAL RETRIEVAL

Our pipeline achieves $\approx 20\%$ Recall@1% and $\approx 50\%$ Recall@5% (corresponding to a Median Rank of 84). This indicates the median distance of each spectrum to the matching scientific summary is 84 among 1,719 candidates, exploring $\approx 5\%$ of the search space. Figure 2 presents the Recall@k% for varying values of $k$.

### 3.2 PHYSICAL INTERPRETATION

Unsurprisingly, autoencoders on spectra are better correlated with the selected physical variables (on average, $|\rho| = 0.43$) than the autoencoders on texts (on average, $|\rho| = 0.30$). Aligning and combining both modalities leads to an even stronger alignment (average $|\rho| = 0.55$). Table 2 shows top correlations: for instance, dimensions $L_{12}$ and $L_1$ encode hardness ratio (hard_hs, $\rho = 0.82$), while $L_{48}$ a captures thermal properties (apec_kt, $\rho \approx 0.74$).

Table 2: Top 10 latent-variable correlations (post-alignment spectra embeddings).

| Latent Dim. | Variable | Correlation |
| --- | --- | --- |
| $L_{12}$ | hard_hs | 0.82 |
| $L_1$ | hard_hs | 0.82 |
| $L_{51}$ | hard_hs | 0.77 |
| $L_{26}$ | hard_hs | 0.76 |
| $L_{48}$ | apec_kt | 0.74 |
| $L_{29}$ | hard_hs | 0.74 |
| $L_{44}$ | hard_hs | 0.71 |
| $L_8$ | powlaw_gamma | 0.68 |
| $L_{30}$ | brems_nh | 0.68 |
| $L_{62}$ | bb_kt | 0.68 |

Table 3: Comparison of Mean Absolute Error (MAE) before and after alignment, obtained using a $k$-NN ($k = 3$) regressor. "Both" corresponds to early fusion by concatenating aligned spectra and text embeddings ($[z_s; z_t] \in \mathbb{R}^{128}$). The "Mean baseline" column reports the MAE obtained when predicting the validation set mean for all samples. The "Improvement" column indicates the relative improvement of the Mixture of Experts (MoE) strategy compared to the best-performing pre-alignment modality. Underlined values represent the lowest MAE (best performance), while bold values indicate results within the confidence interval of the best value.

| Variable | Mean baseline | Pre-alignment | | Post-alignment | | | MoE | Uncertainty | Improvement |
|---|---|---|---|---|---|---|---|---|---|
| | | Spectra | Text | Spectra | Text | Both | | | |
| hard_hs | 0.40 | 0.20 | 0.27 | 0.16 | 0.17 | **0.12** | **0.12** | 0.01 | 40% |
| hard_ms | 0.28 | 0.17 | 0.22 | 0.14 | 0.17 | _0.12_ | _0.12_ | 0.01 | 28% |
| hard_hm | 0.23 | 0.15 | 0.18 | 0.11 | 0.15 | **0.10** | **0.10** | 0.01 | 33% |
| var_prob_b | 0.26 | 0.27 | **0.18** | 0.26 | 0.20 | 0.23 | 0.23 | 0.02 | −25% |
| var_index_b | 1.77 | 1.79 | **1.00** | 1.70 | 1.19 | 1.45 | **1.00** | 0.10 | 0% |
| powlaw_gamma | 0.89 | 0.67 | 0.65 | _0.44_ | 0.67 | **0.41** | **0.41** | 0.05 | 36% |
| powlaw_nh | 65.32 | 33.08 | 48.88 | 26.39 | 40.82 | **21.63** | **21.63** | 2.71 | 35% |
| powlaw_stat | 0.47 | 0.46 | 0.43 | **0.36** | 0.52 | _0.37_ | _0.37_ | 0.04 | 14% |
| bb_kt | 3.23 | 0.32 | 0.49 | **0.27** | 0.40 | _0.28_ | _0.28_ | 0.03 | 13% |
| bb_nh | 58.95 | 74.31 | 111.48 | _53.78_ | 101.59 | **50.29** | **50.29** | 6.30 | 32% |
| bb_stat | 0.91 | 0.75 | 0.78 | **0.65** | 0.79 | _0.68_ | _0.68_ | 0.08 | 10% |
| brems_kt | 11.76 | 6.26 | 6.27 | _5.37_ | **5.35** | 5.64 | 5.64 | 0.67 | 10% |
| brems_nh | 125.98 | 23.31 | 26.98 | _15.57_ | 19.54 | **14.12** | **14.12** | 1.77 | 39% |
| brems_stat | 0.57 | 0.53 | 0.50 | **0.42** | 0.62 | _0.46_ | _0.46_ | 0.05 | 9% |
| apec_kt | 3.51 | 1.60 | 1.74 | _1.23_ | 1.60 | **1.22** | **1.22** | 0.23 | 23% |
| apec_abund | 0.72 | **0.76** | 0.89 | **0.70** | 0.78 | _0.81_ | _0.81_ | 0.13 | −6% |
| apec_z | 0.22 | **0.21** | 0.21 | _0.20_ | 0.21 | _0.20_ | _0.21_ | 0.04 | −3% |
| apec_nh | 65.84 | 35.95 | 35.67 | _27.77_ | 31.47 | **24.92** | **24.92** | 4.65 | 30% |
| apec_stat | 0.73 | _0.66_ | 0.75 | **0.61** | 0.78 | _0.62_ | _0.62_ | 0.11 | 6% |
| flux_significance_b | 8.50 | 7.36 | 7.67 | **4.06** | 6.53 | 4.54 | 4.54 | 0.42 | 38% |

The results also indicate that alignment with scientific papers enhances physical interpretability: the shared latent space better reflects physical variables than either modality alone.

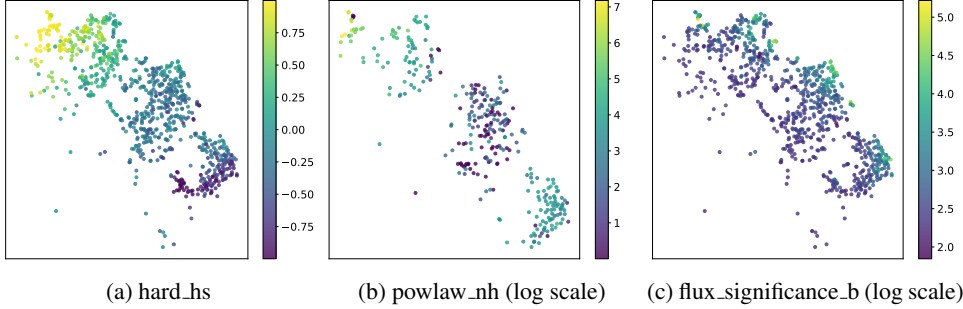

(a) hard_hs     (b) powlaw_nh (log scale)     (c) flux_significance_b (log scale)

Figure 3: tSNE plots for a subset of the physical variables

Beyond individual linear correlations, we analyze whether the latent space is well-structured with respect to the 20 physical variables. To this end, we use $k$-NN to estimate them. Figure 3 presents three tSNE plots that show that multimodal representations generate a well-structured space with respect to a subset of the physical variables. Moreover, multimodal representations substantially improve parameter estimation (Table 3). Using both aligned modalities reduces MAE by $\approx 16\%$ with respect to the best pre-alignment modality, and the MoE strategy further improves to $\approx 18\%$. For hardness ratios (hard_hs, hard_ms, hard_hm), average improvement is 34%. Hydrogen column density ($N_H$) estimates improve by 34% across spectral models. For variability metrics (var_prob_b, var_index_b), text alone performs better because spectral data lacks temporal information, which is lost during alignment.

The 97% compression (4,672 to 128 dimensions[1]) while retaining predictive power is critical for scaling to billion-object surveys (e.g., LSST), where full-dimensionality similarity searches are intractable.

---

[1]64 dimensions in the shared space for each modality

Overall, the emerging structure, not explicitly enforced during training, indicates that contrastive learning preserves domain-relevant information.

## 3.3 OUTLIER DETECTION

Beyond regression and retrieval, the shared latent space enables the discovery of rare astronomical phenomena by identifying points that deviate from the learned multimodal manifold. We apply Isolation Forest, an unsupervised algorithm that isolates outliers by randomly partitioning the high-dimensional feature space, to the aligned concatenated embeddings.

Class-level anomaly-score distributions show that AGNs and AGN candidates have comparable behavior, while QSOs are among the most anomalous well-represented classes. Among less-represented classes, ULXs exhibit high anomaly-score variance, consistent with heterogeneous sub-populations (pulsating and non-pulsating systems). In addition, the relationship between class size and median anomaly score (see also Figure 4) is weakly negative ($\rho \approx -0.14$), indicating that rarity alone does not explain outlierness.

Table 4: Representative top-1% anomalies in the aligned spectra+text latent space (values from the extended analysis).

| Source name | ObsID | Anomaly score | Class |
|---|---|---|---|
| 2CXO J005546.0-721449 | 14671 | 1.0000 | AGN_Candidate |
| 2CXO J004545.5+413942 | 17011 | 0.9746 | GlobCluster |
| 2CXO J234443.9-424312 | 16135 | 0.9310 | Seyfert2 |
| 2CXO J021808.5-051224 | 17301 | 0.8751 | QSO |
| 2CXO J224030.2+032131 | 14516 | 0.8393 | GravLensSystem |
| 2CXO J004722.6-252050 | 3931 | 0.8227 | ULX |

The top 1% anomalies include the gravitational lens system 2CXO J224030.2+032131 and the ULX 2CXO J004722.6-252050. The latter has been independently identified as a candidate pulsating ULX candidate in Pinciroli Vago et al. (2025a), validating the pipeline's ability to discover scientifically interesting objects. It is important to note that the findings in Pinciroli Vago et al. (2025a) were not included in our training dataset, as the publication date succeeded our data collection cut-off. Consequently, the identification of this source as an outlier serves as an independent validation of our model's discovery potential.

## 4 DISCUSSION

Beyond astronomy, this approach applies to any domain with paired observational sequences and textual annotations, such as seismology (which includes waveforms and event reports (Si et al., 2024)), climate science (which includes timeseries and assessment documents (Bai et al., 2024)), and medicine (which comprises physiological signals and clinical notes (Liu et al., 2023)). Our work demonstrates that scientific literature, which is widely available, easy to process, and represents a vast repository of expert knowledge, can be systematically integrated with observational data to create enriched foundation models. This knowledge-augmented paradigm leverages decades of domain expertise, accelerating interpretation and discovery.

For astronomy, this enables:

- Semantic search: given a spectrum, retrieve relevant papers and similar sources, facilitating literature exploration.
- Cross-facility fusion: combine spectra from different instruments (e.g., Chandra, XMM-Newton) via shared textual metadata.
- Automated characterization: preliminary source classification and parameter estimation from latent space, prioritizing follow-up observations.

Moreover, our findings highlight that:

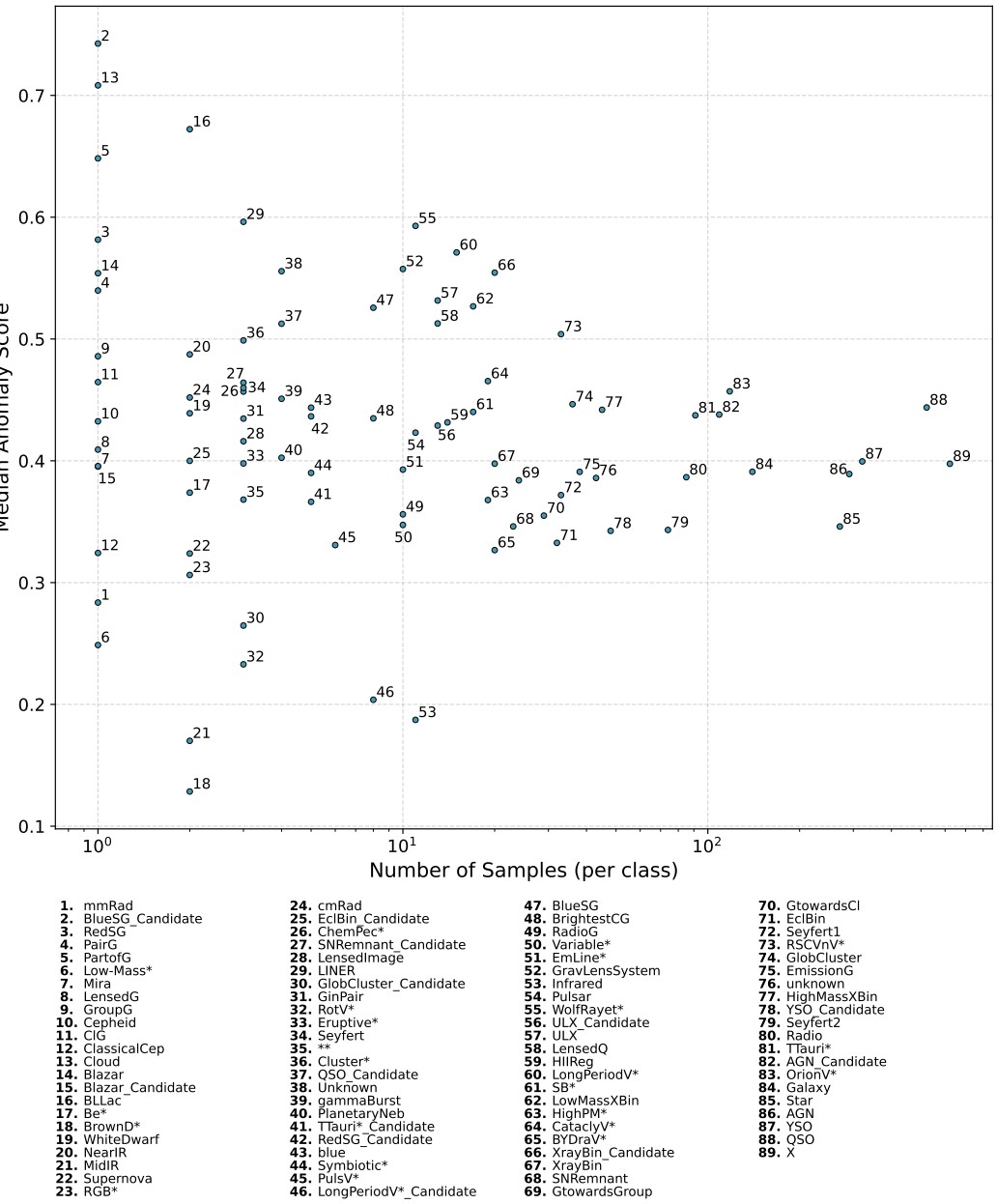

Figure 4: Number of samples and median anomaly score for each class

- Contrastive learning not only enables cross-modal retrieval but also produces latent spaces more correlated with physical variables, an emergent property valuable for scientific applications.
- 97% compression facilitates billion-scale similarity searches, essential for next-generation surveys.

Still, this work has some limitations:

- The retrieval performance of 20% Recall@1% suggests room for improvement. This limitation can be mitigated by improving text summaries and leveraging additional data pairs (possibly adding multimodal observations). Still, the mismatch between X-ray spectra and scientific summaries prevents a perfect alignment.
- Current work focuses on retrieval and regression, and has not proven its usefulness on tasks like text generation from spectra
- Anomaly detection could enhance outlier detection by incorporating physics-based priors, with the goal to prioritize theoretically interesting outliers over statistical artifacts.
- The alignment is limited to astrophysical data, even though the same pipeline can be applied to other fields of science

## 5 CONCLUSIONS

We present the first multimodal foundation model aligning X-ray spectra with scientific texts, demonstrating that knowledge-augmented representation learning can bridge observational data and domain expertise. Key achievements include: (1) 18% improvement in physical parameter estimation via multimodal fusion; (2) 97% data compression preserving correlations with 20 physical variables; (3) discovery of rare phenomena (candidate PULXs, gravitational lenses) through latent space outlier detection.

As petabyte-scale astronomical surveys come online, scalable multimodal approaches are necessary. Our framework, combining self-supervised pre-training, contrastive alignment, and ensemble-based downstream adaptation, offers a blueprint for integrating heterogeneous scientific data. Importantly, this framework can be extended to other scientific domains where aligning observational data with existing literature is possible. By systematically connecting observations with textual knowledge, we move toward foundation models that not only process data efficiently but also encode the semantic richness of scientific understanding.

## DATA AVAILABILITY

The processed data are available at `https://osf.io/56s4h/overview?view_only=1ead08e953814767a6f74e02b73fc40b`. The trained models will be made available upon reasonable request.

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
