# OpenReview forum: "Augmenting representations with scientific papers"
_ICLR.cc/2026/Workshop/FM4Science — ICLR 2026 Workshop FM4Science Poster_

### Official Review · Reviewer_FV5d · 2026-02-14

**Rating:** 6
**Confidence:** 4

**Review:**

## Summary
This paper introduces a contrastive learning framework that aligns X-ray spectra with scientific-text knowledge to build a shared multimodal embedding space. The aligned representations enable spectrum-to-text retrieval (20% Recall@1), improve prediction of 20 physical variables by 16–18% via multimodal fusion and MoE selection, and support outlier detection to surface rare targets such as a candidate pulsating ULX and a gravitational lens system.


## Strength
1. Clear motivation and well-defined problem setting. The paper clearly frames multimodal alignment between spectra and scientific-text descriptions as a way to improve representation learning and enable downstream scientific tasks, and the overall pipeline is easy to follow.

2. Unified framework with multiple downstream evaluations. Beyond the core alignment objective, the work evaluates the learned shared space on several practically relevant tasks—cross-modal retrieval, physical parameter regression, and outlier detection—showing the approach is broadly useful rather than optimized for a single metric.

3. The paper is well written, with clear figures and a coherent presentation that makes the method easy to follow.


## Weakness
1. Unclear fusion of spectra and text representations (Lines 141–144). The paper states that it uses representations from both the spectra and text encoders, but it is not specified how these embeddings are combined for downstream tasks (e.g., concatenation, averaging/summation, or a learnable fusion module). This detail is important for reproducibility and for interpreting the “Both” results in Table 3.

2. The authors use GPT-4o-mini to generate summaries for each scientific papers used for contrastive learning. However in Task 1 CROSS-MODAL RETRIEVAL, the author evaluate the performance on retrieve scientific summary. One question what is the quality of GPT generated summary? I think the authors should manually annotate a small set of image-text pairs, and test on the small human-annotated test set to analyze the summary noise from GPT-4o-mini.

3. The outlier detection section is currently not well supported: it reports class-level trends and highlights specific top-1% anomalies, but provides no figures or tables. Adding at least one visualization and a small table of discovered outliers would make the claims verifiable and more convincing.

4. The authors should clarify whether the trained model will be publicly available to research community for further research use and reproducibility.

---

### Official Review · Reviewer_6N9B · 2026-02-15
**Interesting Idea, Moderate Empirical Depth**

**Rating:** 7
**Confidence:** 4

**Review:**

This paper proposes a multimodal contrastive framework aligning X-ray spectra with scientific paper summaries to create a shared latent space. Spectra are encoded with a transformer-based autoencoder (compressed to 64 dimensions), while paper summaries are embedded via GPT-4o-mini + Ada-002, and both modalities are aligned using InfoNCE loss. The learned space is evaluated on cross-modal retrieval, regression of 20 physical parameters, and anomaly detection. The central claim is that incorporating scientific literature into representation learning improves physical interpretability and downstream scientific tasks.

---

### Meta-Review · Area_Chair_awXP · 2026-02-27

**Recommendation:** Accept (Poster)
**Confidence:** 4

**Metareview:**

Accept.

---

### Decision · Program_Chairs · 2026-03-03

Accept (Poster)